# A Dual Role of Osteopontin in Modifying B Cell Responses

**DOI:** 10.3390/biomedicines11071969

**Published:** 2023-07-12

**Authors:** Rittika Chunder, Verena Schropp, Manuel Marzin, Sandra Amor, Stefanie Kuerten

**Affiliations:** 1Faculty of Medicine, Institute of Neuroanatomy, University of Bonn, 53115 Bonn, Germany; rchunder@uni-bonn.de (R.C.); verena.schropp@uni-bonn.de (V.S.); 2University Hospital Bonn, 53127 Bonn, Germany; 3Institute of Anatomy and Cell Biology, Friedrich-Alexander-Universität Erlangen-Nürnberg, 91054 Erlangen, Germany; 4Department of Pathology, Amsterdam University Medical Center, 1007 MB Amsterdam, The Netherlands; chicho@dds.nl (M.M.); s.amor@amsterdamumc.nl (S.A.); 5Institute of Anatomy, Rostock University Medical Center, 18057 Rostock, Germany

**Keywords:** aggregation, B cells, central nervous system, multiple sclerosis, osteopontin

## Abstract

The occurrence of B cell aggregates within the central nervous system (CNS) has prompted the investigation of the potential sources of pathogenic B cell and T cell responses in a subgroup of secondary progressive multiple sclerosis (MS) patients. Nevertheless, the expression profile of molecules associated with these aggregates and their role in aggregate development and persistence is poorly described. Here, we focused on the expression pattern of osteopontin (OPN), which is a well-described cytokine, in MS brain tissue. Autopsied brain sections from MS cases with and without B cell pathology were screened for the presence of CD20^+^ B cell aggregates and co-expression of OPN. To demonstrate the effect of OPN on B cells, flow cytometry, ELISA and in vitro aggregation assays were conducted using the peripheral blood of healthy volunteers. Although OPN was expressed in MS brain tissue independent of B cell pathology, it was also highly expressed within B cell aggregates. In vitro studies demonstrated that OPN downregulated the co-stimulatory molecules CD80 and CD86 on B cells. OPN-treated B cells produced significantly lower amounts of IL-6. However, OPN-treated B cells also exhibited a higher tendency to form homotypic cell aggregates in vitro. Taken together, our data indicate a conflicting role of OPN in modulating B cell responses.

## 1. Introduction

Multiple sclerosis (MS) is a chronic demyelinating disease of the central nervous system (CNS), which typically manifests in young adults. Although B cells are not the first responders of neuroinflammation, their dichotomous role, especially in the chronic stage of MS has gained increasing attention [1]. Ectopic lymphoid follicles (eLFs), which are a common feature of several chronic autoimmune diseases [2], have been detected in the meninges of some but not all secondary progressive MS patients [3,4,5]. More recent reports have demonstrated the presence of B cell aggregates in other stages of MS [6,7,8]. Furthermore, multiple lines of evidence suggest that the presence of B cell aggregates within the CNS tissue in MS patients positively correlates with a more severe disease pathology [9,10,11].

Nevertheless, there remains a crucial gap in our knowledge regarding the precise molecular and cellular landscape of these aggregates [12]. On the one hand, molecules expressed within the B cell aggregates that may function as positive regulators of disease [13] remain to be clearly defined. On the other hand, the identification of molecules that facilitate B cell aggregation [14,15,16] is an important step to specifically block those pathways that negatively affect the severity of disease progression.

In the current study, we set out to investigate the expression of SPP1—secreted phosphoprotein-1—an extracellular matrix (ECM) glyco-phosphoprotein also described as osteopontin (OPN) [17] within B cell aggregates in MS patients. OPN has been identified as the fifth most common transcript in MS brain tissue [18] and its role in several pathologic and inflammatory processes [19] has been described over the last decades. Furthermore, the involvement of OPN in T cell-mediated CNS pathology has been emphasized in different EAE models [20,21], and very recently, the role of OPN in controlling and compartmentalizing brain T cells has been suggested [22]. While OPN has also been implicated in several diseases with a B cell component [23,24], its expression within eLFs or B cell aggregates in MS patients is unknown.

To this end, we used immunohistochemistry to determine the expression of OPN in MS brain tissue that contained CD20^+^ B cell aggregates. In vitro, human recombinant OPN (rOPN) downregulated the co-stimulatory molecules CD80 and CD86 on human B cells from peripheral blood. Conversely, the stimulation of B cells with rOPN increased the number of B cell aggregates in vitro suggesting that OPN has opposing effects that may have an impact on the pathogenesis of B cell-mediated diseases.

## 2. Materials and Methods

### 2.1. Human Tissue

The MS brain sections used in this study were from the same cohort reported by Chunder et al. [25]. Paraffin-embedded brain tissue sections were received from the Multiple Sclerosis and Parkinson’s Tissue Bank, Center for Brain Sciences, Imperial College London (file number 258/14, approved by the Ethics Committee of the University of Würzburg), or from the Netherlands Brain Bank (NBB; co-ordinator Prof. I. Huitiga) in Amsterdam with approval of the Medical Ethics Committee of the Amsterdam UMC. All participants or the next of kin had given informed consent for an autopsy and the use of tissues for research purposes. MS brain samples from the NBB were selected from regions of interest (ROI) after ex vivo MRI for the detection of lesions. Tissue blocks containing the ROI were cut in half and half fixed in 10% formalin and embedded in paraffin while the other half was snap-frozen and stored in liquid nitrogen. General information on the patient samples is listed in Table 1. *n* = 16 MS patients including *n* = 7 patients with a high B cell pathology were included in the study. Synovial tissues from *n* = 3 patients with rheumatoid arthritis (RA) that were obtained from the Department of Rheumatology and Immunology of University Hospital Erlangen were also included in this study (file number 334_18B, approved by the Ethics Committee of the University Hospital Erlangen).

### 2.2. Immunohistochemistry

The protocol for immunohistochemical staining has been previously described by our group [25]. Briefly, following deparaffinization of the brain sections, heat-induced antigen retrieval was performed in 10 mM sodium citrate buffer (pH 6.0). Sections were washed with tris-buffered saline + 0.05% Tween 20 (TBS-T) between every incubation step. After blocking with 5% milk blocking buffer at room temperature for 1 h, tissue sections were incubated with primary antibodies diluted in 0.5% blocking buffer at 4 °C overnight. The appropriate secondary antibodies diluted in 0.5% blocking buffer were applied to the sections at room temperature for 2 h in the dark. Double stainings were carried out sequentially where the tissue sections were incubated with the second primary antibody for 3 h at room temperature following completion of the first staining. The corresponding secondary antibody was diluted as mentioned above and incubated at room temperature for 1 h. Human brain sections were finally mounted using Fluoroshield mounting medium containing 4′,6-diamidino-2-phenylindole (DAPI) (Abcam, Cambridge, UK). Every staining contained a secondary antibody-only control. Additionally, a rabbit polyclonal IgG control was used to confirm primary antibody binding specificity (Appendix A). The primary and secondary antibodies that were used are listed in Table 2. Images were acquired using a Leica DM6 fluorescence microscope (Leica Microsystems, Wetzlar, Germany) equipped with Las X software (version 5.1.0, Leica Microsystems, Wetzlar, Germany).

### 2.3. B Cell Isolation and Cell Culture 

For the in vitro B cell studies, blood was collected from healthy controls between the ages of 22 and 35. The study was performed pseudonymized and approved by the Ethics Committee of Friedrich-Alexander-Universität Erlangen-Nürnberg, Erlangen, Germany (files 185_18B and 74_18B). Written informed consent was obtained from all volunteers. 

The protocol used for isolating and culturing B cells has been previously described by our group [25]. Briefly, 30 mL of peripheral blood per person was collected using S-Monovette^™^ (Sarstedt Inc., Newton, MA, USA) tubes containing lithium heparin. Whole blood was incubated with RosetteSep^™^ Human B Cell Enrichment Cocktail (STEMCELL Technologies, Vancouver, BC, Canada) and purified by density centrifugation according to the manufacturer’s instructions. Isolated B cells were counted and resuspended in the appropriate B cell medium at a concentration of 150,000 cells/200 μL for all downstream applications. The B cell medium was prepared using Roswell Park Memorial Institute (RPMI)-1640 medium supplemented with 0.3 g/L L-glutamine, 10% fetal bovine serum (FBS) (ThermoFisher, Waltham, MA, USA), 1% penicillin/streptomycin (P/S), 50 μM β-mercaptoethanol, and 10 mM HEPES. An aliquot of 450,000 freshly isolated B cells was set aside for purity check by flow cytometry.

B cells that had a purity of >80% as determined by flow cytometry were cultured in the presence or absence of stimulation (15 ng/mL interleukin (IL)-2 (PeproTech, Rocky Hill, NJ, USA) and 1 μg/mL synthetic TLR 7/8 agonist R-848 (ENZO Life Sciences, Farmingdale, NY, USA)) in 96-well plates. B cells that were stimulated with IL-2 and R-848 are referred to as pre-stimulated B cells, which were incubated at 37 °C and 5% CO_2_ for 24 h before the addition of rOPN. Freshly isolated B cells, which did not receive any stimulation, are mentioned as unstimulated B cells and were directly incubated with rOPN. For every culture condition and all downstream applications, B cells from every donor were plated, measured, and analyzed in duplicates.

### 2.4. Stimulation of B Cells with rOPN

A dose-titrated amount of 2 μg/well (1 μg/mL) human rOPN (R&D Systems, Minneapolis, MN, USA) was added to pre-stimulated or unstimulated B cells, which were cultured for 40 h at 37 °C and 5% CO_2_. The amount of rOPN added to the B cells was optimized to 2 μg/well based on recommendations in the literature [26,27].

### 2.5. Flow Cytometry

#### 2.5.1. Purity Check of B Cells

The purity of the enriched B cell population was measured by flow cytometry as previously described by us [25]. The isolated cell suspensions were incubated with BD Horizon^®^ Fixable Viability Stain 780 (FVS780) (BD Biosciences, Franklin Lakes, NJ, USA) at 4 °C for 30 min in the dark. Cells were then washed, centrifuged, and incubated with anti-human anti-CD19 and anti-CD45 antibodies at 4 °C in the dark for 30 min. Following a washing step, cells were resuspended in an appropriate volume of FACSFlow^™^ (BD Biosciences) for measurement of the samples. Data were acquired on a CytoFLEX S flow cytometer (Beckman Coulter, Brea, CA, USA) equipped with CytExpert software (version 2.2, Beckman Coulter, Brea, CA, USA), and analysis was performed with FlowJo software (version 10.8.0, BD Biosciences, Franklin Lakes, NJ, USA). A table corresponding to the B cell purities from the different donors used in this study is shown in Appendix A.

#### 2.5.2. B Cell Activation

To check the activation status of pre-stimulated or unstimulated B cells that were treated with human rOPN for 40 h, the following cell surface markers were used: CD3, CD20, CD80, and CD86. Live cells were discriminated from dead ones using FVS780. Staining was performed as described above. The gating strategy used for the measurement of B cell activation status has been previously described by our group [25] and is shown in Appendix A. Unstained, isotype staining controls and fluorescence minus one (FMO) samples were included.

Cases where either rOPN-treated or untreated B cells showed a coefficient of variation (CV) higher than 20% between the technical replicates for any condition were excluded from the analysis. Materials and methods with an overlap between the current manuscript and that reported in Chunder et al. [25] have been summarized in Appendix A.

### 2.6. Antibody Array and ELISA

To investigate the cytokine secretion profile of pre-stimulated B cells after incubation with rOPN, a human cytokine antibody array kit with 23 targets (Abcam) was used (Appendix A). An equal volume of culture supernatant from either rOPN-treated or untreated pre-stimulated B cells was used for the array and the assay was performed according to the manufacturer’s instructions. The integrated density of the dot blots was analyzed using Image J (NIH). Based on the preliminary findings using the antibody array, quantitative analysis of IL-6 and IL-10 was performed on the B cell culture supernatant. Both ELISAs were performed as per the manufacturer’s protocol (ThermoFisher, Waltham, MA, USA).

### 2.7. Aggregation Assay

Aggregation assays using purified B cells were performed as described earlier [28]. Briefly, pre-stimulated B cells that were treated with rOPN for 40 h were incubated with mouse anti-human CD19 at a concentration of 1 μg/mL for 2 h at 37 °C and 5% CO_2_. Mouse anti-human IgG1 was used as an isotype control antibody. Five images were acquired from each well using a Leica DMIL LED microscope (Leica Microsystems, Wetzlar, Germany). B cell aggregation was quantified with Fiji (version 2.3.1, National Institutes of Health (NIH), Bethesda, MD, USA), with at least three images/well included in the analysis. Images/wells where a bubble or the edge of the well was photographed were excluded from the analysis.

### 2.8. Statistical Analysis

Statistical analysis was performed using GraphPad Prism (version 8.0, GraphPad Software Inc., San Diego, CA, USA). A Shapiro–Wilk normality test was used to test the Gaussian distribution of every dataset. Differences between parametric groups were assessed using either a *t* test or a one-way ANOVA, while for non-parametric datasets, the Wilcoxon test was used. The level of significance was set at 5%.

## 3. Results

### 3.1. OPN Is Expressed within B Cell Aggregates in Human Tissues

We checked for the expression pattern of OPN in MS brain sections from a cohort of patients that had already been characterized by our group [25]. As expected, the expression of OPN was not limited to sections with a prominent B cell pathology (Figure 1A). Tissue sections from patients that were positive only for T cells (CD3^+^ staining) and those without any CD3/CD20 staining (“No T cell and B cell” infiltration) also expressed high amounts of OPN. However, the results also demonstrated that this molecule is strongly expressed within CD20^+^ B cell aggregates (Figure 1B). Furthermore, we confirmed previous findings [29] using synovial tissue from RA patients to demonstrate that OPN is also expressed within B cell aggregates in the periphery (Figure 1C). Control stainings are shown in Appendix A.

### 3.2. B Cells Treated with OPN Downregulate the Co-Stimulatory Molecules CD80 and CD86

Having shown that OPN is expressed within B cell aggregates, we wanted to investigate whether it had any effect on the activation status of B cells. To demonstrate the influence of OPN on B cells, we studied the expression pattern of CD80 and CD86 as two co-stimulatory molecules that are expressed on activated B cells and also play a crucial role in T cell activation [30,31]. Unstimulated peripheral blood B cells from *n* = 6 healthy donors treated with human rOPN for 40 h in culture significantly downregulated CD80 (*p* = 0.027) as determined by flow cytometry (Figure 2A). Flow cytometric analysis of B cells that were first pre-stimulated with R-848 and IL-2 and then treated with rOPN (*n* = 6 healthy donors) demonstrated a significantly lower expression of both co-stimulatory molecules (*p* = 0.003 for CD80; *p* = 0.010 for CD86) (Figure 2B) compared to the vehicle (i.e., medium or R-848 + IL-2 only, respectively).

### 3.3. OPN Has Differential Effects on IL-6 and IL-10 Secretion by B Cells

To investigate if rOPN also had an effect on the cytokine profile of B cells, a dot blot antibody array was performed. Supernatants of B cells that had been pre-stimulated with R-848 and IL-2 only or pre-stimulated and then treated with rOPN from *n* = 1 donor were used. Based on the integrated density of the dots (Appendix A), two of the cytokines that triggered our interest were IL-10 and IL-6, which showed increased or decreased secretion by rOPN-treated B cells, respectively (Figure 3A), compared to the vehicle. An IL-6/IL-10-producing B cell ratio to predict active disease in MS patients has been studied earlier [32]. Although no statistics could be performed on the dot blot with *n* = 1 patient, we verified the influence of rOPN on IL-6 and IL-10 secretion by B cells using ELISA. A supernatant of pre-stimulated B cells (of >80% purity) treated with rOPN from a cohort of *n* = 6 healthy donors was tested for both cytokines. There was significant downregulation in the synthesis of IL-6 (*p* = 0.020) in rOPN-treated B cells vs. the untreated group, but no significance was achieved in IL-10 production between the groups.

An estimation plot was created to document the concurrent IL-6 and IL-10 synthesis profile of B cells treated with rOPN from each individual donor. Although no statistical significance was observed (*p* = 0.062), rOPN-treated B cells from 5/6 donors that had decreased levels of IL-6 produced an increased amount of IL-10 (Figure 3C), while this was not the case in the group that was treated with R-848 and IL-2 alone.

### 3.4. OPN Significantly Increases the Potential of B Cells to Form Aggregates

Given the expression of OPN within B cell aggregates, we assessed the impact of rOPN on homotypic B cell aggregation. Accordingly, we used the in vitro B cell aggregation assay model initially described by Smith, Rigley, and Callard [33] (Figure 4A). The number of B cell aggregates induced by anti-CD19 antibody in rOPN-treated B cells was significantly higher (*p* = 0.019) than in the control group (Figure 4B). However, the size of the aggregates formed was comparable between the two groups.

## 4. Discussion

In this study, we show that OPN is expressed within B cell aggregates in MS brain tissue and uncover a conflicting role of this molecule in its association with B cells. On the one hand, OPN has the potential to dampen B cell activity by downregulating the co-stimulatory molecules CD80 and CD86. On the other hand, B cells have an increased tendency to form aggregates when treated with OPN.

First, we demonstrate strong expression of OPN within B cell aggregates in human brain tissues. Using MS brain tissues that were positive for B cell aggregates, we extend previous findings that OPN expression is associated with B cell clusters not only in peripheral inflammatory autoimmune diseases [34,35] but also in the event of chronic neuroinflammation.

In autoimmune diseases, OPN is widely considered to be a pro-inflammatory cytokine that enhances the production of IL-6 [29] and (auto)antibodies [24] by B cells. However, OPN also has anti-inflammatory properties and is known to regulate tissue repair [36]. Our study demonstrates that OPN has the potential to regulate the activation status of B cells and therefore possibly also plays an important role in modulating T cell activation.

CD27^+^ memory B cells are not only aberrantly outnumbered in the peripheral blood and cerebrospinal fluid (CSF) of MS patients [37] but have also been detected within MS lesions [7]. Furthermore, it has been demonstrated that memory B cells can function as potent antigen-presenting cells (APCs) [38] with pro-inflammatory tendencies [39] and are the main pathogenic subset of B cells [40]. Therefore, given the relevance of memory B cells in MS, we investigated the effects of OPN on B cells stimulated with a combination of R-848 and IL-2 as polyclonal activators that prompt efficient proliferation and differentiation of memory B cells [41].

Our data indicate that OPN significantly downregulated both CD80 and CD86 on B cells pre-stimulated with R-848 and IL-2 while only affecting CD80 expression on unstimulated B cells. CD80 and CD86 are co-stimulatory molecules that are overexpressed on B cells in MS patients [42] and are involved in immune regulation [43]. Future studies focusing on a more comprehensive phenotype characterization will allow us to confirm the nature of the effect of OPN on different subtypes of B cells. Nevertheless, our results suggest that OPN reduces the overall activation status of B cells through modulation of its co-stimulatory molecules CD80/CD86.

Additionally, on the one hand, we show a significant downregulation of IL-6 secretion by pre-stimulated B cells treated with rOPN and an upregulation of IL-10 synthesis by the same cells in 83.3% of our cohort. On the other hand, as expected, we also confirm that pre-stimulated B cells that were left untreated concurrently secreted high levels of IL-6 and low levels of IL-10 in all six donors tested. Despite the small sample size, the data strongly support an overall tendency of OPN to polarize B cells to a more anti-inflammatory phenotype. The therapeutic benefit of neutralizing the effects of the pro-inflammatory cytokine IL-6 has been suggested for different autoimmune diseases such as RA [44,45], and harvesting the potential of B cell-derived IL-10 in controlling autoimmunity, such as in MS, has also been widely discussed [46,47].

Of note, in contrast to our finding, other groups have suggested that OPN not only inhibits IL-10 secretion in macrophages [48] but OPN-deficient mice produce an enhanced level of IL-10 compared to wild-type mice [49]. This suggests that OPN most likely exerts differential effects on distinct cell populations in a context-dependent manner. One caveat of this study is that we cannot exclude the possibility of a small number of IL-6/IL-10 secreting contaminating cell populations in our B cell culture that contributed to the signal detected by the different ELISAs. Therefore, taking the heterogeneity of the immune repertoires between donors into account, in vitro treatment of B cells with rOPN using a larger cohort of donors would verify the beneficial effect of this molecule in limiting inflammation. Furthermore, the choice of cytokines used to demonstrate the anti-inflammatory effect of rOPN on B cells (i.e., IL-6 and IL-10) was shortlisted from a cohort of only *n* = 1 donor. Hence, a more comprehensive screening of pro- and anti-inflammatory cytokines secreted by OPN-stimulated B cells using a larger cohort of donors would strengthen our current findings.

Finally, the high level of OPN expression within the B cell aggregates in the human brain sections of MS patients led us to question whether OPN also influenced the formation of B cell aggregates. Surprisingly, B cells treated with rOPN had a significantly higher tendency to form aggregates than the control. An earlier study, however, argued against the nature of OPN to induce homotypic aggregation formation [50]. Using CD44-transfected cell lines, Weber et al. showed that the interaction between OPN and its receptor CD44 did not induce homotypic aggregation [50]. Despite the conflicting findings, it is plausible that in our in vitro system, OPN-stimulated B cells have a higher tendency to form aggregates that may be dependent on another receptor–ligand interaction. Identifying the precise receptor–ligand interaction, which results in OPN-dependent B cell aggregation, would be a necessary step in the future.

One of the limitations of our short-term in vitro assay is that it does not mimic the chronic aggregation of B cells. Furthermore, B cell aggregates found in chronically inflamed tissues, such as in MS or RA, do not contain a pure population of B cells but are rather clusters of B cells with T cells, as shown by us and others [27,51]. Therefore, future studies using a mixed lymphocyte population to observe the effect of OPN on facilitating B cell aggregation longitudinally would validate the role of this molecule in the development of aggregates in chronic neuroinflammation.

## 5. Conclusions

To conclude, the pleiotropic nature of OPN is well known [52] given its capacity to interact with multiple receptors [53]. Exploiting the interaction between OPN and its receptors that results in polarizing B cells to a less activated phenotype could open up new therapeutic possibilities in chronic (neuro)inflammatory diseases. Similarly, inhibiting the interaction between OPN and possible putative ligands that enhance B cell aggregate formation could limit an ongoing antigen-specific immune response within tissues.

## Figures and Tables

**Figure 1 biomedicines-11-01969-f001:**
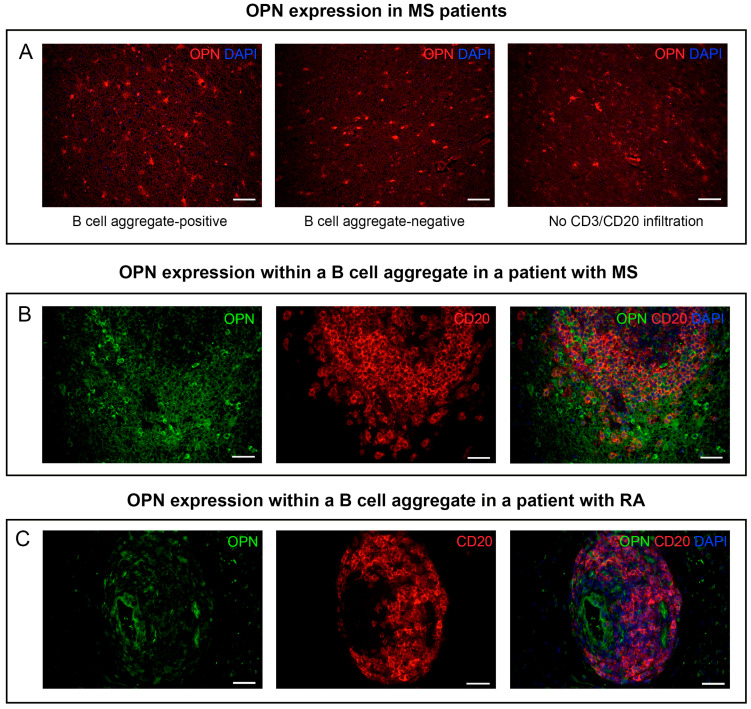
Expression of OPN in human tissue sections with or without B cell pathology. (**A**) OPN expression in brain tissue from a CD20^+^ B cell aggregate-positive patient, a CD20^+^ B cell aggregate-negative patient, and a patient without any observable CD3/CD20 infiltration. Expression of OPN within a CD20^+^ B cell aggregate in a (**B**) patient with MS vs. (**C**) a patient with RA. Scale bars represent either 50 μm (**A** and **B**) or 100 μm (**C**). MS = multiple sclerosis; OPN = osteopontin; RA = rheumatoid arthritis.

**Figure 2 biomedicines-11-01969-f002:**
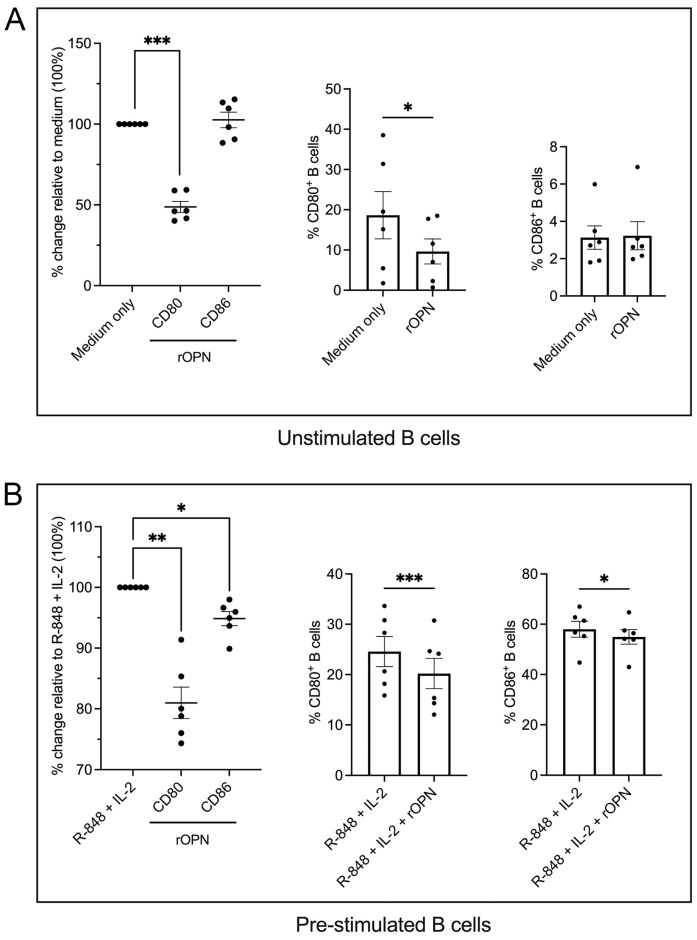
Analysis of co-stimulatory molecules on B cells after rOPN stimulation. (**A**) Expression of CD80 and CD86 by unstimulated B cells following incubation with medium only or rOPN (cells isolated from *n* = 6 healthy donors). (**B**) Expression of CD80 and CD86 by B cells pre-stimulated with R-848 + IL-2 and incubated with or without rOPN (cells isolated from *n* = 6 healthy donors). Every data point on the graphs corresponds to the percentage of either CD80^+^ or CD86^+^ B cells under different conditions. Graphs show mean values ± standard error of mean (SEM). * *p* < 0.05, ** *p* < 0.01, *** *p* < 0.001, paired *t* test. IL = interleukin; R-848 = synthetic toll-like receptor 7/8 agonist; rOPN = recombinant osteopontin.

**Figure 3 biomedicines-11-01969-f003:**
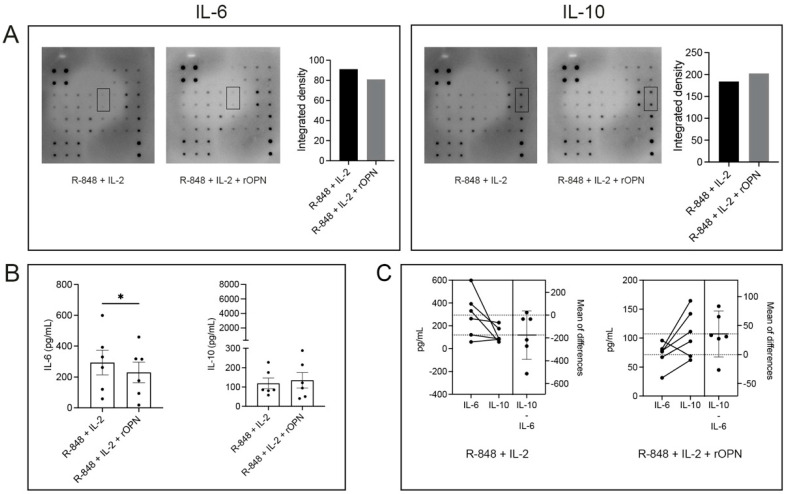
Cytokine profiling of rOPN-treated B cells. (**A**) IL-6 and IL-10 production by B cells from the same donor either with or without treatment with rOPN is shown. The amount of IL-6 and IL-10 production between the two groups was calculated by using the integrated density from *n* = 1 donor, measured in duplicates as shown in the black boxes. (**B**) IL-6 and IL-10 production of pre-stimulated B cells (*n* = 6 healthy donors), measured by ELISA. (**C**) An estimation plot demonstrating the level of IL-6 vs. IL-10 secretion by rOPN-treated and -untreated B cells are also shown. Bars display mean values ± SEM. * *p* < 0.05, paired *t* test. IL = interleukin; R-848 = synthetic toll-like receptor 7/8 agonist; rOPN = recombinant osteopontin.

**Figure 4 biomedicines-11-01969-f004:**
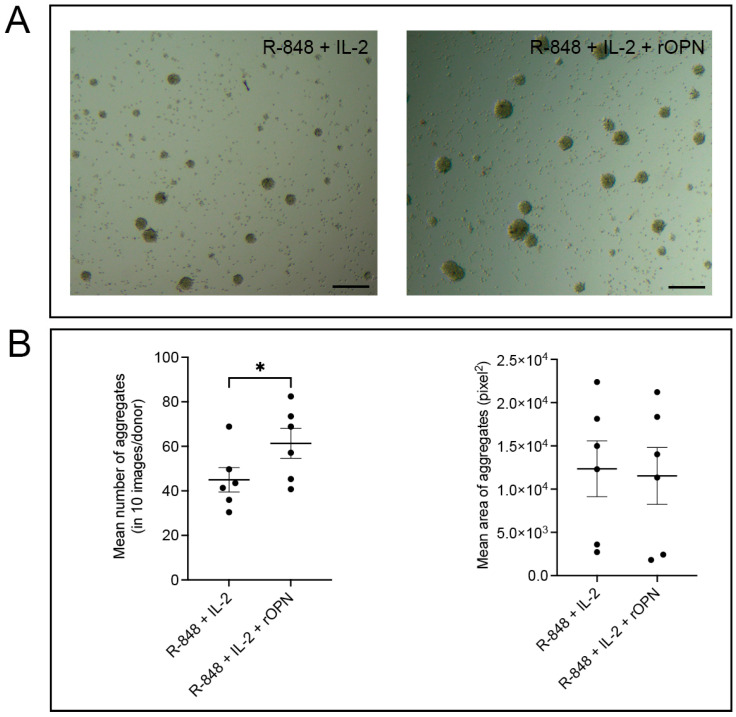
Aggregation of B cells in the presence of rOPN. (**A**) Representative images of aggregation of pre-stimulated B cells with or without rOPN treatment induced by anti-CD19 antibody. Scale bars represent 100 μm. (**B**) Number and size of the aggregates formed comparing control (R-848 + IL-2 stimulation only) and rOPN (in addition to R-848 + IL-2)-treated B cells. Bars display mean values ± SEM. * *p* < 0.05, paired *t* test. IL = interleukin; R-848 = synthetic toll-like receptor 7/8 agonist; rOPN = recombinant osteopontin.

**Table 1 biomedicines-11-01969-t001:** Patient details [25].

**MS patients**							
**Number**	**Age**	**Sex**	**MS Type**	**Disease Duration** **(In Years)**	**Cause of Death**	**Number of Blocks**	**Patient Type**
MS325	51	M	PPMS	2	Bronchopneumonia	2	NI
MS342	35	F	SPMS	5	MS	2	B cell
MS402	46	M	SPMS	20	MS, bronchopneumonia	1	B cell
MS407	44	F	SPMS	19	Sepsis, pneumonia	1	B cell
MS408	39	M	SPMS	10	Sepsis, pneumonia	1	NI
MS438	53	F	Unknown	18	MS	2	NI
MS444	49	M	SPMS	20	Kidney failure	2	T cell
MS473	39	F	PPMS	13	MS, bronchopneumonia	2	T cell
MS485	57	F	PPMS	29	MS, bronchopneumonia	2	T cell
MS510	38	F	SPMS	22	MS, pneumonia	1	T cell
MS523	63	F	SPMS	32	Bronchopneumonia	1	T cell
MS528	45	F	SPMS	25	Pneumonia	1	T cell
11-077	66	F	PPMS	32	Euthanasia	5	B cell
14-006	56	F	Unknown	31	Suicide by pentobarbital intoxication	2	B cell
15-047	50	F	SPMS	18	Euthanasia	2	B cell
16-019	48	M	Unknown	15	Ileus, dehydration	4	B cell
**RA patients**							
**Number**	**Age**		**Sex**	**Tissue type**		**Number of blocks**	
427-19	57		F	Surgical biopsy		1	
470-19a	74		F	Surgical biopsy		1	
471-19b	82		F	Surgical biopsy		1	

F = female; M = male; MS = multiple sclerosis; NI= No CD3/CD20 infiltration; PPMS = primary progressive MS; SPMS = secondary progressive MS.

**Table 2 biomedicines-11-01969-t002:** List of antibodies and their dilutions.

Antibodies Used for Immunofluorescence				
Antibody	Source	Host Species	Clone	Dilution
Anti-CD20	Invitrogen	Mouse	L26	1:400
Anti-CD3	Abcam	Rabbit	SP162	1:150
Anti-OPN			Polyclonal	1:100
Anti-human IgG	Abcam	Rabbit	Polyclonal	1:200
**Antibodies used for flow cytometry**				
BB700 Anti-CD3	BD Biosciences	Mouse	SK7	1:20
APC Anti-CD19			HIB19	1:5
APC Anti-CD20			2H7	1:5
BB515 Anti-CD45			HI30	1:20
BV605 Anti-CD80			2D10.4	1:40
BB515 Anti-CD86			FUN-1	1:20

## Data Availability

The original contributions presented in the study are included in the article/Appendix A, and further inquiries can be directed to the corresponding author.

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
