# Peer review of "A Dual Role of Osteopontin in Modifying B Cell Responses"

_biomedicines, 2023, doi:10.3390/biomedicines11071969_

Round 1

Reviewer 1 Report

This is a study that examines the role of osteopontin in multiple sclerosis. The authors show expression in MS lesions and that osteopontin inhibits B-71 expression while promoting IL-10 production and B cell aggregation. The authors conclude that osteopontin has conflicting effects in MS.

The authors should note the initial Science publication by Steinman's group which showed increased IL-10 production in osteopontin-deficient mice. It suggests that the context in which osteopontin is present can have differential effects. Even though promoting B cell aggregation, one might argue that the effects shown in this study are mostly inhibitory and that increasing the number of B cells with reduced B7-1 expression and increased IL-10 expression may have an overall inhibitory effect. 

Author Response

This is a study that examines the role of osteopontin in multiple sclerosis. The authors show expression in MS lesions and that osteopontin inhibits B-71 expression while promoting IL-10 production and B cell aggregation. The authors conclude that osteopontin has conflicting effects in MS.

The authors should note the initial Science publication by Steinman's group which showed increased IL-10 production in osteopontin-deficient mice. It suggests that the context in which osteopontin is present can have differential effects. Even though promoting B cell aggregation, one might argue that the effects shown in this study are mostly inhibitory and that increasing the number of B cells with reduced B7-1 expression and increased IL-10 expression may have an overall inhibitory effect. 

We thank the reviewer for this comment. We have now integrated the Science paper in the revised manuscript (please refer to Lines 314 – 320).

Reviewer 2 Report

In the manuscript of Chunder et al., a dual role of osteopontin in modifying B cell responses is suggested.

Some recent reports demonstrated, at least a part of MS patients, the presence of  B cell aggregates in the CNS tissue .

In brain tissue of MS patients osteopontin is highly expressed  also in B cell aggregates. Nevertheless a polyclonal antibody is used, which might cause a strong background. The isotype control with rabbit IgG is missing. In the periphery (here in synovial tissue of RA patient) osteopontin seems to be highly expressed (in CD20+ cells? )

What is shown in Figure 2? Are that results of flow cytometry? It is not mentioned in the manuscript. How were the results calculated?. This is also not mentioned. (also not in material and method)

The supplementary file 1 shows the gating strategy of flow cytometry of B cells. Where is the isotype control? How were CD80 positive B cells and CD86 positive B cells gated? This is not possible to understand if no negative control is  available, since the possibly positive populations are no clear separated populations.

In Figure 3 the B cells from peripheral blood of healthy donors were isolated. The purity is not described and shown. The purified B cells are positive (how many percent of which markers?) for CD20? In the manuscript only the enrichment of  B cells is described. Cells were stimulated with IL-2 and recombinant osteopontin. How do the authors know, that only B cells are isolated and proliferating? After doing an array with the conditioned media of the cultured cells IL-6 and IL-10 seemed to be differntly expressed. Quantification with ELISA show a significant difference in the reduction of IL-6 and an increased amount of IL-10 (there is almost no increase!)

What is the putative ligand of osteopontin to form aggregates of B cells?

All in all, what do the data mean?

Author Response

Chunder et al. investigated the effects of osteopontin on B cells in the context of multiple sclerosis (MS). Osteopontin was found in the brains of people with progressive forms of MS within and outside of B cell aggregates.  Recombinant osteopontin down-regulated CD80 and CD86 on B cells in vitro. Osteopontin-treated B cells produced less interleukin-6 (IL-6).

We thank the reviewer for the positive evaluation of the manuscript and for the suggestions to strengthen the manuscript. We hope to have addressed all suggestions and comments.

1) Figure 1. A: Please consider changing the labels "B cell patient" and "T cell patient." There is no "B cell patient" to a "T cell patient." Please use different terminology to explain.

We have now replaced “B cell patient” and “T cell patient” with a new terminology (CD20+ B cell aggregate-positive or CD20+cell aggregate-negative patient, respectively) which we hope fits better. Please refer to Figure 1 of the revised manuscript.

2) Figure 3. A. The authors have measured many cytokines (23 "targets") but only provide integrated density measures for IL-6 and IL-10. Can such quantification be shown for the other cytokines? If not, please justify your answer (e.g. low abundance making quantification unreliable). 

We thank the reviewer for this comment. Indeed, the quantification was difficult with all the targets and because of the limited number of samples (n = 1). However, we have now added a Supplementary Table S2 including the integrated densities for the other cytokines from the dot blot array that could be quantified using ImageJ. We have made a note of this in the revised manuscript (Lines 323 – 327).

3) Figure 1 A and B have quite some "background staining." Would it be possible to address this?  

We have tried to modify Figures 1A and B to reduce the “background staining”. Furthermore, we have now added a rabbit polyclonal IgG control to confirm the specificity of the OPN staining. Please refer to Additional File 1, Supplementary Figure S1.

4) DISCUSSION. Please discuss the differences between B cell aggregation in the CNS of people with MS vs. B cell aggregation in vitro. What are the limitations of the in vivo approach? How are in vivo aggregates different in terms of their cellular components (i.e. is it really just B cells?) and other molecules/pathways that may play a role in aggregate formation in vivo that are impossible or difficult to mimic in vitro? What about the chronicity of the B cell aggregates in vivo? i.e. can chronic aggregation of B cells in follicles be mimicked in a short-term culture?

We thank the reviewer for this very important point. We have now modified the Discussion to include this caveat of our assay (i.e., the difference between chronic inflammation and the short-term in vitro B cell aggregation assay). Please refer to Lines 339 – 345 of the revised manuscript.

Reviewer 3 Report

Chunder et al. investigated the effects of osteopontin on B cells in the context of multiple sclerosis (MS). Osteopontin was found in the brains of people with progressive forms of MS within and outside of B cell aggregates.  Recombinant osteopontin down-regulated CD80 and CD86 on B cells in vitro. Osteopontin-treated B cells produced less interleukin-6 (IL-6).

I would like to suggest addressing the following points to strengthen the manuscript further:

Major

1. Figure 1. A: Please consider changing the labels "B cell patient" and "T cell patient." There is no "B cell patient" to a "T cell patient." Please use different terminology to explain.

2. Figure 3. A. The authors have measured many cytokines (23 "targets") but only provide integrated density measures for IL-6 and IL-10. Can such quantification be shown for the other cytokines? If not, please justify your answer (e.g. low abundance making quantification unreliable). 

Minor

1. Figure 1 A and B have quite some "background staining." Would it be possible to address this?  

2. DISCUSSION. Please discuss the differences between B cell aggregation in the CNS of people with MS vs. B cell aggregation in vitro. What are the limitations of the in vivo approach? How are in vivo aggregates different in terms of their cellular components (i.e. is it really just B cells?) and other molecules/pathways that may play a role in aggregate formation in vivo that are impossible or difficult to mimic in vitro? What about the chronicity of the B cell aggregates in vivo? I.e. can chronic aggregation of B cells in follicles be mimicked in a short-term culture?

Author Response

As the authors state, OPN is a highly pleiotropic cytokine that has the capacity to interact with multiple receptors and can often have conflicting functions in individual tissues.  Here the authors suggest OPN has an overall anti-inflammatory role on B cell function in MS where it decreases CD80 and CD86, and IL6 - but can increase B cell aggregate formation.  This reviewer has only minor observations and suggestions.

We thank the reviewer for the suggestions and for taking the time to evaluate the manuscript. We have tried to address the questions to the best of our ability.

1) Throughout the text when citing figures do so using the A, B, C et designations ie., Figure 1A rather than Figure 1.

Please refer to our revised manuscript where we have now cited the figures more precisely as suggested by the reviewer.

2) The secondary antibody control is not a proper control for IHC.  IHC requires a non-relevant primary IgG control.  Please see Hewitt et al., J histochem & Cytochem 2014;62:693 for explanation.

We thank the reviewer for this suggestion and have now included an appropriate antibody control for the OPN antibody which is a rabbit polyclonal. Please refer to Additional File 1, Supplementary Figure S1.

3) Line 121 - How did you arrive at the 20 ug/well dose?

We are thankful to the reviewer for pointing this out and we apologize that we had incorrectly mentioned the 20 mg/well dose. In fact, the dose should have been a 2 mg/well (concentration of 1 mg/mL) which we have now corrected. We sincerely apologize for this mistake. We have made the corresponding corrections in the revised manuscript (please refer to Lines 131 – 134).

4) Table 2 - Please include the sources of the antibodies.

We have now added the sources of the antibodies. Please refer to the revised manuscript, Table 2.

5) Figure 1 - you do not represent controls in your figures.  These need to be shown. In the legend, please mention CD20.

We have now added the secondary only controls for the CD20/OPN staining in the Additional file 1 (Supplementary Figure S1). Of note, given that the MS brain tissue sections with CD20+ B cell aggregates are extremely valuable given the scarcity of receiving them and in addition to the limited number of slides that we had received/have, we have used B cell containing synovial tissue from RA patients for the double staining with secondary antibodies to OPN and CD20.

6) Lines 186-189 - This is all said in a confusing way.  State somewhere that they are expressed on activated B cells.

We apologize for the confusion and have now re-phrased the sentences (Lines 212 – 214, revised manuscript).

7) Lines 207-209 - I don't see any statistics indicating a statistical increase or decrease.

We clarified this in the revised manuscript (Lines 240 – 242). Please also refer to Lines 323 – 327 for further clarification.

8) Lines 217-219 - Same as comment (7).

We have now added the relevant statistics in the revised manuscript (Lines 247 – 248).

9) Lines 290-295 - I think this requires a more detailed discussion because it is potentially damaging to the study if not argued properly.

We thank the reviewer for pointing this out and have now added a more detailed explanation (Lines 339 – 345) in the revised manuscript.

Reviewer 4 Report

As the authors state, OPN is a highly pleiotropic cytokine that has the capacity to interact with multiple receptors and can often have conflicting functions in individual tissues.  Here the authors suggest OPN has an overall anti-inflammatory role on B cell function in MS where it decreases CD80 and CD86, and IL6 - but can increase B cell aggregate formation.  This reviewer has only minor observations and suggestions:

1) Throughout the text when citing figures do so using the A, B, C et designations ie., Figure 1A rather than Figure 1.

2) The secondary antibody control is not a proper control for IHC.  IHC requires a non-relevant primary IgG control.  Please see Hewitt et al., J histochem & Cytochem 2014;62:693 for explanation.

3) Line 121 - How did you arrive at the 20 ug/well dose?

4) Table 2 - Please include the sources of the antibodies.

5) Figure 1 - you do not represent controls in your figures.  These need to be shown. In the legend please mention CD20.

6) Lines 186-189 - This is all said in a confusing way.  State somewhere that they are expressed on activated B cells.

7) Lines 207-209 - I don't see any statistics indicating a statistical increase or decrease.

8) Lines 217-219 - Same as comment (7).

9) Lines 290-295 - I think this requires a more detailed discussion because it is potentially damaging to the study if not argued properly.

Author Response

1) In the manuscript of Chunder et al., a dual role of osteopontin in modifying B cell responses is suggested. Some recent reports demonstrated, at least a part of MS patients, the presence of  B cell aggregates in the CNS tissue. In brain tissue of MS patients osteopontin is highly expressed also in B cell aggregates.

Nevertheless a polyclonal antibody is used, which might cause a strong background. The isotype control with rabbit IgG is missing. In the periphery (here in synovial tissue of RA patient) osteopontin seems to be highly expressed (in CD20+ cells?)

We thank the reviewer for raising this concern and have now included an appropriate antibody control for the OPN antibody which is a rabbit polyclonal. Please refer to Additional File 1, Supplementary Figure S1. Indeed we saw some CD20+ OPN+ cells in the periphery.

2) What is shown in Figure 2? Are that results of flow cytometry? It is not mentioned in the manuscript. How were the results calculated?. This is also not mentioned. (also not in material and method)

We apologize for this lack of clarity. We have now mentioned flow cytometry used as a read-out method in the results section (revised version Lines 215 – 220).

3) The supplementary file 1 shows the gating strategy of flow cytometry of B cells. Where is the isotype control? How were CD80 positive B cells and CD86 positive B cells gated? This is not possible to understand if no negative control is available, since the possibly positive populations are no clear separated populations.

We thank the reviewer for this question. We have now modified our supplementary figure xx in the Additional file 1 to include the isotype controls. We have modified the figure legend accordingly.

We would like to clarify that Supplementary Figure S2 is very similar to Figure S2 from our previously published manuscript (Chunder, Schropp, Jabari, Marzin, Amor, Kuerten, 2022; https://www.frontiersin.org/articles/10.3389/fimmu.2022.1025377/full). We had started out working on rOPN and MMP-3 (matrix metalloproteinase-3) in parallel and used almost identical culture conditions, flow cytometry protocols and optimizations. First we did a purity check of the B cells and determined the total count of the purified B cells acquired from every donor. In the event that we had sufficient number of isolated B cells, we divided the cells from the same donor and stimulated them with either rMMP-3 or rOPN. Therefore, there is a significant overlap in the gating strategies and cell culture methods, etc.

We have now added an additional figure (Supplementary Figure S3) with the list of materials and methods which have an overlap between the two manuscripts.

4) In Figure 3 the B cells from peripheral blood of healthy donors were isolated. The purity is not described and shown. The purified B cells are positive (how many percent of which markers?) for CD20? In the manuscript only the enrichment of  B cells is described.

Thank you for this important question. Indeed we checked for purity of the B cells by flow cytometry (Materials and Methods, section 2.5.1) and have now included a table corresponding to the purity of the cells in Additional file 1 (Supplementary Table S1).

5) Cells were stimulated with IL-2 and recombinant osteopontin. How do the authors know, that only B cells are isolated and proliferating? After doing an array with the conditioned media of the cultured cells IL-6 and IL-10 seemed to be differntly expressed. Quantification with ELISA show a significant difference in the reduction of IL-6 and an increased amount of IL-10 (there is almost no increase!)

We understand the reviewer’s concern regarding the specificity of the assay. On the one hand, we only used supernatants of B cell culture that were > 80% pure for the ELISA assays. On the other hand, we also confirmed for purity of the B cells after stimulation by gating this population during our analysis. Since we included CD3 as marker to account for T cell contamination in the “B cell activation panel”, we were able to verify the % of CD20+ CD3- B cells in our culture. Indeed, it is possible that some of the signal in the ELISAs comes from other cell populations and have now added this caveat in the discussion (Lines 317 – 320).

6) What is the putative ligand of osteopontin to form aggregates of B cells?

As described by Weber et al. in 1996, CD44, which is expressed on B cells, is a receptor for osteopontin. However, the authors did not observe any homotypic aggregation in their transfected cells. OPN is known to interact with multiple other ligands/receptors. Identifying the precise receptor-ligand interaction which results in OPN dependent B cell aggregation is a work we aim to do in the future. Please refer to Lines 335 – 337 of the revised manuscript.

7) All in all, what do the data mean?

Thank you for this question. Along with the changes made in the revised manuscript, we hope that we are able to convince the reviewer that OPN has the potential to play both a positive and a negative role in (neuro)inflammation. Our data validates the pleiotropic nature of OPN in the context of MS and that the influence of OPN on B cells is dependent on its interaction with a specific receptor/ligand.  Please refer to Lines 348 – 353 of the revised manuscript which summarizes the findings of our data.

Round 2

Reviewer 2 Report

The authors answered all questions and showed controls . The clarity of figure 2 war improved.

Please note in the conclusions that 

....Similarly, inhibiting the interaction between OPN and POSSIBLE PUTATIVE ligandS that enhances B cell....

Otherwise it is misleading. 

Author Response

We would like to thank the reviewer for the remaining comment. We have changed the sentence in the revised manuscript accordingly. 

Reviewer 3 Report

I have no further comments

Author Response

There are no specific questions to address. We thank the reviewer for the positive evaluation.